# CCFont: Component-Based Chinese Font Generation Model Using Generative Adversarial Networks (GANs)

Jangkyoung Park [ID], Ammar Ul Hassan and Jaeyoung Choi *[ID]

School of Computer Science and Engineering, Soongsil University, Seoul 06978, Korea
* Correspondence: choi@ssu.ac.kr

**Abstract:** Font generation using deep learning has made considerable progress using image style transfer, but the automatic conversion/generation of Chinese characters still remains a difficult task owing to the complex character shape and large number of Chinese characters. Most known Chinese character generation models use the image conversion method of the Chinese character shape itself; however, it is difficult to reproduce complex Chinese characters. Recent methods have utilized character compositionality by separating up to three or four components to improve the quality of generated characters, but it is still difficult to generate high-quality results for complex Chinese characters with many components. In this study, we proposed the CCFont model (component-based Chinese font generation model using generative adversarial networks (GANs)) that automatically generates all Chinese characters using Chinese character components (up to 17 components). The CCFont model generates all Chinese characters in various styles using the components of Chinese characters based on conditional GAN. By acquiring local style information from the components, the information is more accurate and there is less information loss than when global information is obtained from the image of the entire character, reducing the failure of style conversion and improving quality to produce high-quality results. Additionally, the CCFont model generates high-quality results without any additional training (zero-shot font generation without any additional training) for the first-seen characters and styles. For example, the CCFont model, which was trained with only traditional Chinese (TC) characters, generates high-quality results for languages that can be divided into components, such as Korean and Thai, as well as simplified Chinese (SC) characters that are only seen during inference. CCFont can be adopted as a multi-lingual font-generation model that can be applied to all languages, which can be divided into components. To the best of our knowledge, the proposed method is the first to generate a zero-shot multilingual generation model using components. Qualitative and quantitative experiments were conducted to demonstrate the effectiveness of the proposed method.

**Keywords:** Chinese font generation; radicals/components; GAN

## 1. Introduction

It is a well-known fact that creating new fonts using deep learning is very efficient in terms of time and labor cost. When a font designer creates a new set of English fonts, they only need to design 52 uppercase and lowercase Roman characters. However, it is necessary to design 11,172 characters for Hangul and 70,000 to 100,000 characters for Chinese, which is labor-intensive and practically impossible. These labor-intensive tasks take around 700 days for Hangul and more than 10 years for Chinese, and it is almost impossible to design them in the same style, even if a designer makes one character every 30 min and works 8 h a day [1].

With deep learning based on artificial intelligence, if you design only 256 characters, it is now possible to create a new set of Korean fonts in 30 min, and this can also be applied to Chinese characters. Therefore, interest in developing a model that automatically produces

fonts for various languages has recently increased, and active research is being conducted on the generation of various characters, including Korean, Chinese, and English [2–5].

Most font generation models use the generative adversarial network (GAN) [6,7] as a basic frame, and various modified forms are used to achieve great results. However, in the case of Hangul and Hanja, which have a large number of characters and complex shapes, it is not easy to change their character style. This is because it is difficult to convert an image by acquiring style information from the content. In particular, it is difficult to successfully convert content images into complex Chinese characters [8–10]. Figure 1 shows a sample failure case for the generation of Chinese characters.

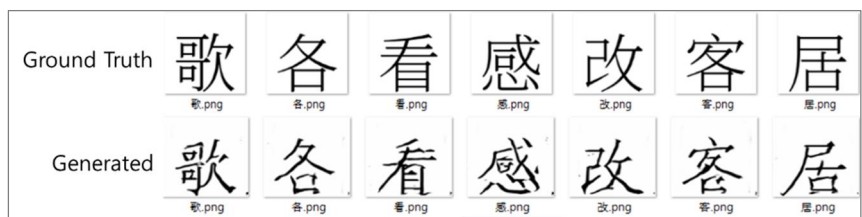

**Figure 1.** A sample case of failures to generate Chinese characters.

To solve this problem, SKFont [11] proposed a three-stage network based on character image skeletonization [12] for generating Hangul font images. Additionally, the direct conversion of relatively simple components has also been attempted [13–21].

Recently, a component-based method for Hangul was introduced that generated high-quality results [20]. In Hangul, characters can be regenerated with a maximum of three components. In contrast, in Chinese characters, the number of components varies for each character, and many exceed 10 components. Therefore, a new model must be created to apply the component concepts to Chinese characters.

In this study, we proposed a component-based Chinese font generation (CCFont) model that further separates Chinese character radicals into components and automatically generates high-quality Chinese characters using all components. The CCFont is trained to separate all the components constituting Chinese characters from the 2003 traditional Chinese (TC) characters currently used in China and combines them in a new style. The CCFont model automatically generates not only almost all TC characters (max 17 components) but also all 7445 simplified Chinese (SC) characters of GB/T 2312-1980, which includes 99.99% of the Chinese characters currently used in China, in real time with a high-quality new style.

Because our proposed method obtains style information from the components and not from the glyphs, the model reduces the number of failure cases. It operates in an end-to-end manner that converts styles using component images of Chinese characters and can generate high-quality Chinese characters without additional training steps for unseen font styles.

The CCFont model is executed step by step as follows:

1. Data generation (Character-to-Radical-to-Components Module, CRC-Module);
2. Model training (CCFont Training Module);
3. Generating new character/style Chinese characters (CCFont Generating Chinese Characters, GC-Module).

The data generation module breaks Chinese characters into components to generate data for the CCFont Training Module, which receives the data and regenerates the images through training. For generating a new character, the GC-Module uses the CCFont training module to generate Chinese characters. Figure 2 depicts an example of Chinese characters generated (out) by the CCFont model in the style of the target character (tgt) based on character input (src).

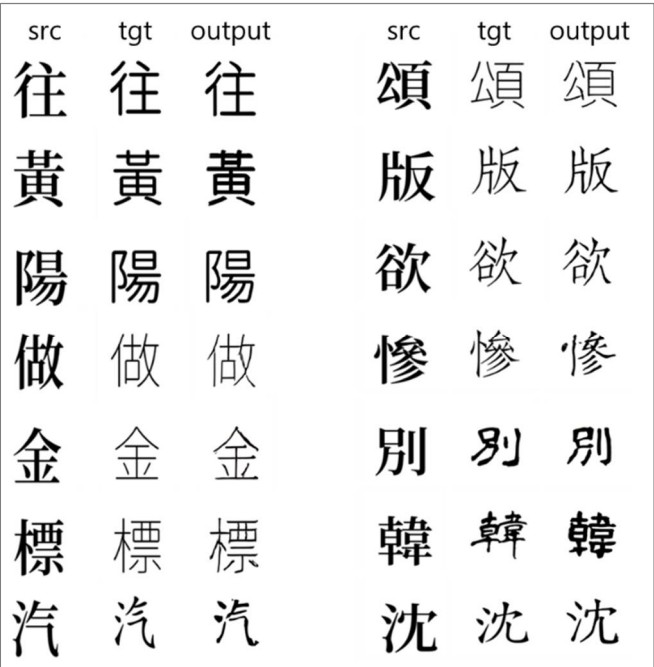

**Figure 2.** Example of Chinese characters generated (out) by CCFont model in the style of the target character (tgt) based on the character input (src).

The CCFont model proposed in this study has the following advantages and contributions:

- Decomposing Chinese characters into components decreases the rendering difficulties significantly.
- Component image conversion, which is much simpler than character shapes, significantly reduces the number of failure cases and enables high-quality character generation.
- Up to 17 components can be generated, making it possible to generate almost all Chinese characters (99.99% in use).
- The development of TC and SC Chinese character automatic generation/con-version model (CCFont) using Chinese character components.
- Experiments demonstrated zero-shot font generation and generated Korean (Hangul) and Thai fonts using the CCFont model.

There are two types of Chinese characters, TC and SC. In this paper, we trained our model with only TC characters and demonstrated that the model is generalized to generate the unseen SC characters. Additionally, we conducted experiments to demonstrate that the proposed method can also generate multi-lingual characters such as Korean Hangul and Thai characters.

## 2. Related Works

The style conversion of Chinese fonts remains a challenging task because of the large number of characters (70,224 characters, GB18010-2005) and complex shapes (some characters have more than 50 strokes) compared to Roman characters. Several models based on GAN [6,7] have recently been proposed and have demonstrated success in font synthesis tasks [8–12,22–26]. We divided these methods into two categories, many-shot and few-shot font generation methods, which are discussed in the next section.

### 2.1. Many-Shot Font Generation Methods

Most of these methods consider font generation as an image-to-image translation [27] problem based on conditional GAN (cGAN) [28]. The conditional GAN is the extension of vanilla GAN, where the image is generated with some condition c, i.e., c can be a class label or an image (as in our case). The condition c is added to both the generator and dis-

criminator for the parameters intended to be controlled. Rewrite [29] proposed a top-down CNN architecture for generating one font style. Zi2zi [8] extended the one-to-one mapping function of the pix2pix [27] framework by proposing a one-to-many mapping function that utilizes category embedding for style injection. Based on zi2zi, DCFont [9] used a separate style feature extractor to generate Chinese handwriting characters. SCFont [10] extracts skeletons/strokes from source glyphs and converts them into target styles. Ko et al. [12] proposed a method for font image skeletonization based on the pix2pix framework. Later, this method was extended as a three-stage stack network architecture for generating Korean Hangul fonts [11]. Some methods have also utilized Chinese character radicals for Chinese character recognition [13,14]. Wen et al. [16] generated handwritten Chinese font characters by refining strokes of Chinese characters. More recently, RCN [18] was proposed to generate new Chinese character categories by integrating radicals. All of these methods produce satisfactory results; however, these methods are unable to generate unseen font styles without additional finetuning steps on a large number of reference characters. For example, zi2zi [8], DCFont [9], and SCFont [10] require more than 700 reference characters of unseen font to learn the new style, which is time consuming and computationally expensive. To overcome these issues, recent methods have focused on generating characters in a few-shot setting.

### 2.2. Few-Shot-Learning

In a few-shot setting, the goal is to generate characters in an unseen font style with just a few reference characters of that style at the test time. Recent few-shot methods have tackled this font image generation by utilizing the composition of characters. For example, DM-Font [24] generates Hangul and Thai characters with very few reference characters. It extracts the component information of Hangul characters and stores it in memory and uses it whenever necessary. This takes a significant amount of time, and only Hangul stored in memory can be generated during the inference time. Another problem with DM-Font is that it cannot generate other language characters, such as Chinese. By improving the problem of DM-Font, LF-Font [25], which enables Chinese characters, uses a style encoder and a content encoder for each component to generate Korean and Chinese characters without fine-tuning. However, due to architecture constraints, LF-Font is unable to perform multi-lingual font style transfer.

To overcome LF-Font issues, MX-Font [26] proposed multilingual font generation architecture by utilizing multi-head encoders for each reference image to separate information between content and style, enabling work across languages. It has superior performance to DM-Font and LF-Font and produces high-quality results; however, training a character for the first time requires repeated exposure. Recently, RD-GAN [17] was proposed based on Chinese character radicals for zero shot Chinese character style transfer based on radical decompositions. However, RD-GAN is unable to produce unseen font styles.

More recently, a fine-grained local style from reference style characters is extracted [19]. However, because of the complex structure, such as more than 200 components of Chinese characters, only experimental results using certain components were presented, and limited performance was shown for complex characters. CKFont [20] pairs the source character with the individual components of the target character, obtains style information from the individual components of the target character, and generates results. As a result, it is possible to expand and apply not only Hangul but also Chinese and Thai characters, as it generates all the characters by training to separate and combine components, and it produces high-quality results in a short time.

Obtaining component images from a character image is extremely limited and challenging when the character's shape is complex. We resolved this issue by separating the component from the shape of the character and then recombining the transformed component image by converting the separated component image, yielding positive results. Style information and content structure are the most important factors in glyph style transfer. Obtaining accurate style information and continuously transmitting information without

any loss are critical for image reproduction. To overcome such style information acquisition and loss, we developed the proposed CCFont model that separates the complex shapes of Chinese characters into components (decomposition), extracts local styles from these components separately, and generates characters of the target style by recombining the extracted component (composition) features. The trained model, in this way, reconstructs the characters in the introduced method, regardless of the content of the components. Therefore, it worked successfully for all Chinese characters (TC and SC) that could be separated into components, as well as Hangul and Thai characters. In other words, the CCFont model is a new multi-lingual (or cross-lingual) font generation model [26] of the zero-shot concept [15,17] that works even with characters and styles that the model sees for the first time.

## 3. Structure of Chinese Character

### 3.1. Character-Sets and Strokes (https://en.wikipedia.org/wiki/Chinese_characters, (accessed on 1 June 2022))

The exact number of Chinese characters is unknown because anyone can create characters by combining sounds and meanings. In record, there are 47,035 characters in the 1716 Kangxi Zidian (康熙字典) dictionary. The 1989 Hanyu Da Zidian (漢語大字典) dictionary has 54,678 characters, the 1994 Zhonghua Zihai (中华字海) has 85,568, and the 2004 Yitizi Zidian (異体字字典) contains 106,230 characters. The Japanese 2003 Dai Kan-Wa Jiten (大漢和辞典), which uses Chinese characters, contains 50,305 characters, and Korea's 2008 Han-Han Dae Sajeon (漢韓大辭典) contains 53,667 characters.

Owing to the complexity of Chinese characters, it is difficult not only to understand all Chinese characters but also to write them; therefore, SC characters have been used recently. SC is a simplified version of the number of strokes and shapes compared to the original TC.

GB2312-1980, published in 1980 by the Chinese government, contains 6763 SC Hanzi (汉字), currently used in mainland China and Singapore. In 1995, this was extended to GBK, including TC (20,914 characters), GBK21003 (21,003 characters), GBK26634 (26,634 characters), and GB18030 (70,244 characters), which were combined with GB2312. In 2013, the Chinese government classified the number of Chinese characters used into three grades according to the difficulty of Chinese characters and published 8105 standard characters including SC (Tōngyòng Guīfàn Hànzì Biǎo, 通用规范汉字表). There are 3500 characters in Level 1, 3000 characters in Level 2, and Level 3 has 1605 characters. Furthermore, Levels 1 and 2 are regulated by common Chinese characters.

Chinese characters can be divided into strokes and their components (radicals). A stroke is the smallest unit comprising a character. There are six types of basic strokes, as shown in Figure 3, and a total of 41 types of strokes.

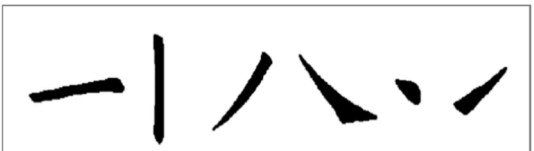

**Figure 3.** Basic six strokes of Chinese characters.

The strokes form part of a character but have no meaning. The strokes are useful for writing and identifying characters, but they are too small to be used as elements to form an image of a character. Figure 4 shows examples of a large number of strokes and the complexity of Chinese characters. The characters with the most strokes currently in use are the 58 characters for BiangBiang noodles, a type of noodle from the Chinese province of Shaanxi. There are also characters with higher strokes (64, 84, etc.) to denote the complexity of Chinese characters, but they are seldom used.

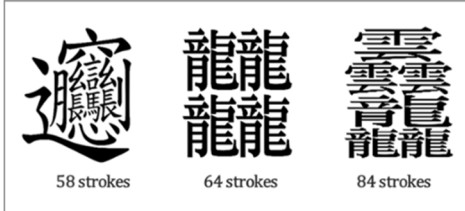

**Figure 4.** Examples of the complexity of Chinese characters.

*3.2. Radicals and Components (http://hanzidb.org/, (accessed on 1 June 2022))*

To find a Chinese character in a dictionary, you need an index of Chinese characters, similar to the Roman alphabet; this is called a radical (piānpáng, 偏旁) or indexing component (部首) [20]. The exact number of radicals is unknown and depends on the number of characters in the character set. There are 214 Kangxi radicals used in China today and 188 in the *Oxford Concise English Chinese Dictionary* (ISBN 0-19-596457-8). The standard GF 0011-2009 (汉字部首表) has 201 radicals in simplified Chinese characters. The number of Chinese characters commonly used today is approximately 3500, and there are approximately 500 unique radicals [17].

Radicals are divided into phonetic and meaning radicals. For example, as shown in Figure 5 (left), the TC character 媽 mā "mother" in the left part is the radical 女 n ǚ "female"—the semantic component—and the right part 馬 mǎ "horse" is the phonetic component. In Figure 5, the left side is traditional, the right side is simplified, and the same rule applies to both.

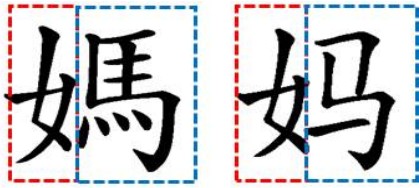

**Figure 5.** The radials are separated to the left and right of the character.

In addition, the same radical may change in shape, as shown in Figure 6, depending on the location where it is used 心 = 忄, 手 = 扌 = 才, 火 = 灬, etc. Because the radical shape changes depending on character position, it is more difficult to change the style of Chinese characters. We solved this problem by using the image separated into the target character components and by using the modified radical image as is.

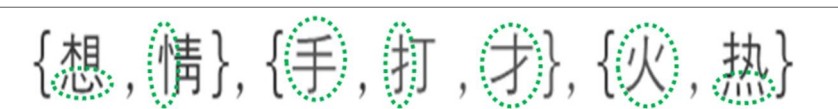

**Figure 6.** Examples of different shapes of the same components.

Radicals can again be separated into parts (components), which are then divided into lower subcomponents and divided again until they become basic components that cannot be further divided. For example, as shown in Figure 7, a character can be divided into one to two radicals and 1 to 14 components until it cannot be divided further.

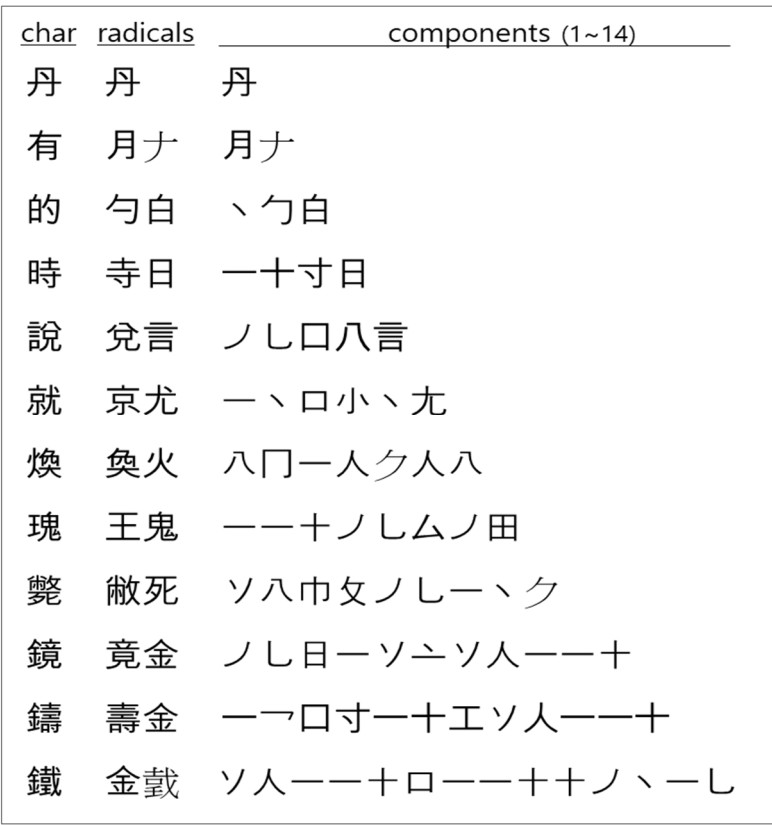

**Figure 7.** Examples of components from a character.

Basic components are sometimes used repeatedly such as 多, 晶, etc., and the basic components themselves become characters such as 言, 心, etc.

## 4. Training Methodology

In the case of a Chinese character with a complex shape, it is difficult to convert the style by mapping the character itself; therefore, we used a method of decomposing the character into components, transforming the decomposed components into styles, and then recombining the transformed components to generate a new style of character. This is different from [13–15,17–19], which separated character images into component images. These models cannot separate component images in the case of complex characters.

Figure 8 shows the process of decomposing the characters into components and composing the characters from the decomposed components. For example, separating the character of '説' gives compositions of ['兌', '言'] (level 1), and by separating again '兌' and '言' (level 2) until they can no longer be separated (level 4), 説 is finally divided into five basic components of ['丿', '乚', '口', '八', '言'] (decomposition). When the new style of '説' is generated, the newly styled basic components are composed in the opposite direction (composition).

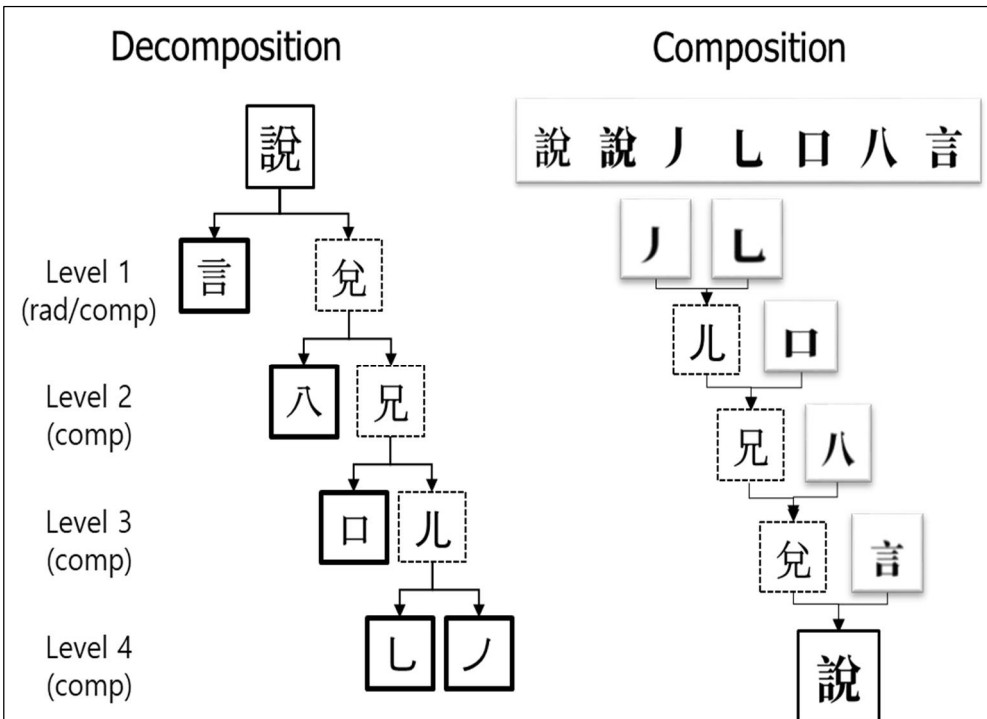

**Figure 8.** An example of character decomposition and composition.

The sample input image to the CCFont model is shown in Figure 9, where src represents the source image, tgt represents the target image, and all the basic components of the tgt image representing the target font style (tgt style-wised components) are demonstrated.

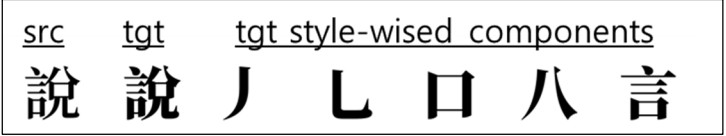

**Figure 9.** An example of input images. Src and tgt represents source and target images, respectively.

The components were provided separately in Python as 'compositions' of the 'CJKradlib RadicalFinder' API (https://pypi.org/project/cjkradlib/, (accessed on 1 June 2022)). We decomposed and composed Python programs using API.

Most Chinese characters have fewer than 14 components (99.99%), but some have more than 14 components. In addition, SC character components had fewer than TC components. In the examples in Figure 10, the character 鐵 (iron), which is the character with the greatest number of components among GB2312, is divided into 14 components, as shown in Figure 10a, and is reduced to four components, shown in Figure 10b, when divided into SC characters. SC radicals significantly reduced the number of components by converting traditional components into simple and non-separable components, i.e., 言 -> 讠, 金 -> 钅, 食 -> 饣, etc. Figure 10c shows the TC characteristics with 15, 16, and 17 components, which are the three characters with the highest number of components in GBK21003 (21,003 characters).

**Figure 10.** (**a**) Decomposition examples of 14 components of 鐵 in TC; (**b**) four components of 铁 in SC; and (**c**) three-character examples of more than 14 components in GBK21003.

According to the Chinese character set, the number of characters, radicals, and components vary, as does the number of characters and radicals. Table 1 lists the number of unique radicals and components for each set of characters. It demonstrates the variation in character shape, number of characters based on the character set, number of radicals and components, and minimum number of characters necessary to obtain the component. As shown in Table 1, GB2312 contained 281 unique components, GBK21003 contained 347 unique components, and GBK26634 contained 346 unique components.

**Table 1.** Radicals and components of character sets.

| Char-Set | Char Type | Total Chars | Unique Radicals | Unique Components | Min No. of Char for Unique Components |
|---|---|---|---|---|---|
| Common use 2003 chars | TC | 2003 | 603 | 249 | 212 |
| Common use 3500 chars | TC | 3500 | 747 | 267 | 250 |
| GB2312 | SC | 6763 | 1081 | 281 | 242 |
| GBK21003 | TC | 21,003 | 2174 | 347 | 325 |
| GBK26634 | TC | 26,634 | 2136 | 346 | 307 |

In this study, 2000 commercial Chinese characters from standard Chinese GB2312 (up to 14 components) and three characters with 15, 16, and 17 components from GBK21003 were added. Therefore, the CCFont model was trained with 2003 TC characters, 603 unique radicals, and 249 basic components, covering up to 17 components (99.99% of common TC/SC). In other words, the model can regenerate all Chinese characters with 17 components or fewer. Therefore, the proposed model works for any character that can be separated into components (in less than 17s). As a result, it applies to Hangul (three components) and Thai (four components), as well as to TC/SC characters used in China, Japan, and Korea (CJK).

The ability to generate multiple languages with the CCFont model trained with only TC characters demonstrates that the computer can generate new characters through deep learning by executing recombination regardless of the component contents.

## 5. CCFont Model

### 5.1. Model Architecture

To generate high-quality converted images from complex-shaped Chinese characters, it is necessary to accurately extract target-style information features such that no information is lost during learning.

As shown in Figure 11, we used two encoders for this purpose, which modified a general GAN structure, one to generate font content information (Ec) and the other to generate target font style information (Es). This was intended to obtain more precise style information from the target font components, and the image for each component was maintained to prevent loss.

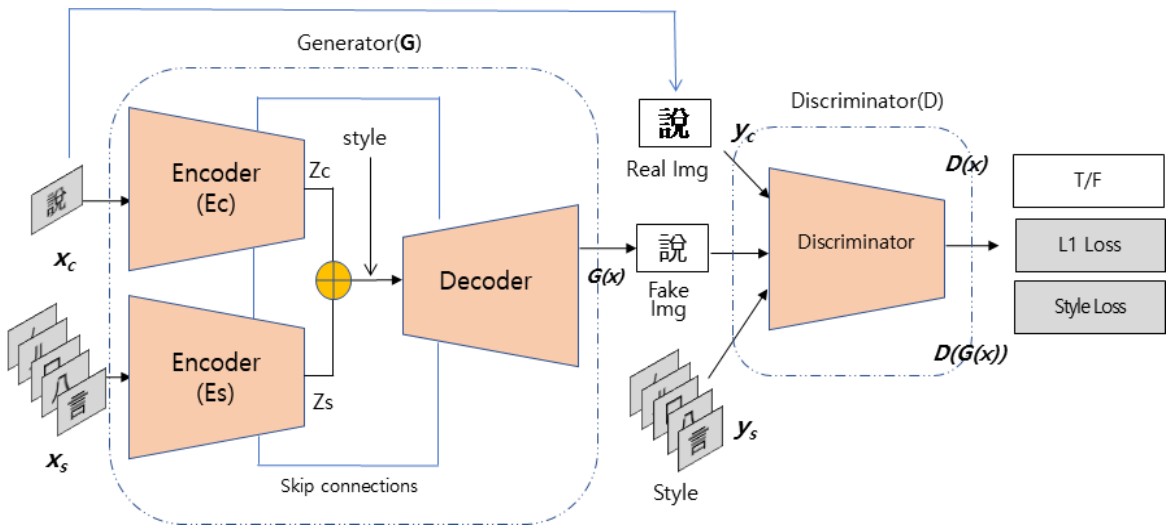

**Figure 11.** CCFont model architecture.

As shown in Figure 11, the CCFont is a concise model that utilizes the composition and components of Chinese characters in a one-to-many framework. It is a conditional GAN [30] structure that uses the content image (Xc) and the style image (Xs) of the character component to be converted as input and has two encoder structures (Ec, Es) to input them separately using the concatenated vector.

The encoder Ec/Es downsamples the input image into eight layers. Each convolution layer uses kernel = 4 and stride = 2, followed by instance normalization, with the exception of the first layer, which uses kernel = 7 and stride = 1. We used LeakyRelu as the activation function, and all the layers had a skip connection structure.

The resulting latent vectors from the two encoders, Zc and Zs, are merged and used as inputs to the decoder and then upsampled again through eight convolutional layers to generate content in a transformed style. Each layer undergoes deconvolution using kernel = 5, stride = 1, instance normalization, and ReLU functions. All layers have a skip connection structure, in which the encoder layer and content/style are merged with the upsampled vectors. Finally, the resulting image G(x) is generated through convolution and tanh functions. The generated result is input to discriminator D, and the GAN loss is calculated by comparing the source image and the generated result image. Character and style loss is achieved by comparing the vector generated from the source image with the resulting image vector, and the quality of the image is improved by comparing the target image and the generated image (L1 loss).

*5.2. Loss Functions*

The total loss function L of the model is expressed as the sum of adversarial loss ($L_{ADV}$), style loss ($L_s$), and L1 loss ($L_{L1}$), as shown in Equation (1). $\lambda_s$ and $\lambda_{L1}$ are hyperparameters for style and L1 loss during training and serve as weights for each loss.

$$L = arg\ min_G\ max_D\ L_{ADV}\ (G, D) + \lambda_s\ L_S\ (G, D) + \lambda_{L1}\ L_{L1}(G) \tag{1}$$

The loss functions in Equation (1) are as follows:

**Adversarial Loss ($L_{ADV}$):** The $L_{ADV}$ of cGAN is expressed as Equation (2), known as min-max adversarial loss [6], where $G$ minimizes (generator loss) and $D$ maximizes (discriminator loss). The discriminator $D$ determines whether the received fake image ($G(x)$) is real (true: 1) or fake (false: 0) and maximizes it ($D(y)$ = 1). Generator G generates a fake image ($G(x)$) and forces D to predict it as a real image ($D(G(x))$ = 1), such that $G(x)$ is minimal ($G(x)$ = 0).

$$min_G\ max_D\ L_{ADV}\ (G, D) = E_y\ [log\ D(y)] + E_x\ [log\ (1 - D(G(x)))], \tag{2}$$

where $y$ is the real image of the character that corresponds to $Yc$, and $x$ equals $Zc + Zs$. By predicting $y$ as a real image and $x$ as a fake image from $G$, the discriminator $D$ attempts to minimize this loss function. In comparison, the generator $G$ attempts to maximize $x$ as a true image to deceive the discriminator $D$.

**Style Classification Loss ($Ls$):** To create a one-to-many style-converted image, $D$ determines whether the style is the same as the target style while discriminating the real from the fake style ($D(y_S)$). To maintain the font style of the image, $D$ predicts the font style (0–1) and feeds it back to $G$ to reduce loss and lets $G$ generate a font with the correct style. This concept is the same as adversarial loss, and it applies to character and is the same as Equation (3), which maximizes $D(y_S)$ and minimizes $G(x_S)$.

$$min_G\ max_D\ L_S\ (G, D) = Ey\ [log\ D(y_S)] + Ex\ [log\ (1 - D(G(x_S)))] \tag{3}$$

**L1 Loss ($L_{L1}$):** $L_{L1}$ makes the two images equal to $G$, generating a fake image ($G(x)$) and reducing the mean absolute error (MAE) compared with the pixel in the target image ($Y$), which can be expressed as Equation (4):

$$L_{L1} = E_{x,\ y}\ [||y - G(x)||] \tag{4}$$

## 6. Experiments and Results

*6.1. Data Set Preparation (Character-to-Radical-to-Components)*

To obtain the basic components of a Chinese character, as depicted in Figure 6, all characters must be decomposed to level N (for example, N = 12 for 17 components), and to convert the components into the target style, the target character must be decomposed into the basic components in advance.

To create a training dataset, 2000 random Chinese characters were extracted from the Chinese national standard GB2312 (6763 characters, including 99.99% Chinese characters), along with 15 (霽), 16 (鬱), and 17 (钃) component characters from GBK21003 (21,003 characters) were added, and 19 font styles were used for training. We used 2003 Chinese characters with a maximum of 17 components and 19 Chinese character font styles.

By using Python CJKradlib RadicalFinder's API, all characters could be separated into basic components (up to 17 components), and the style of the separated components is extracted through the component images of the target character style and combined with the character content (source character). The images of resolution $256 \times 256 \times 19$ ($256 \times 4864$) with total number of 38,057 images ($2003 \times 19$) and respective component images 160,341 were prepared.

For font style, 19 styles that generally work for all characters were extracted among the many font styles, and the font list is as follows; 19 styled example characters are shown in Figure 12.

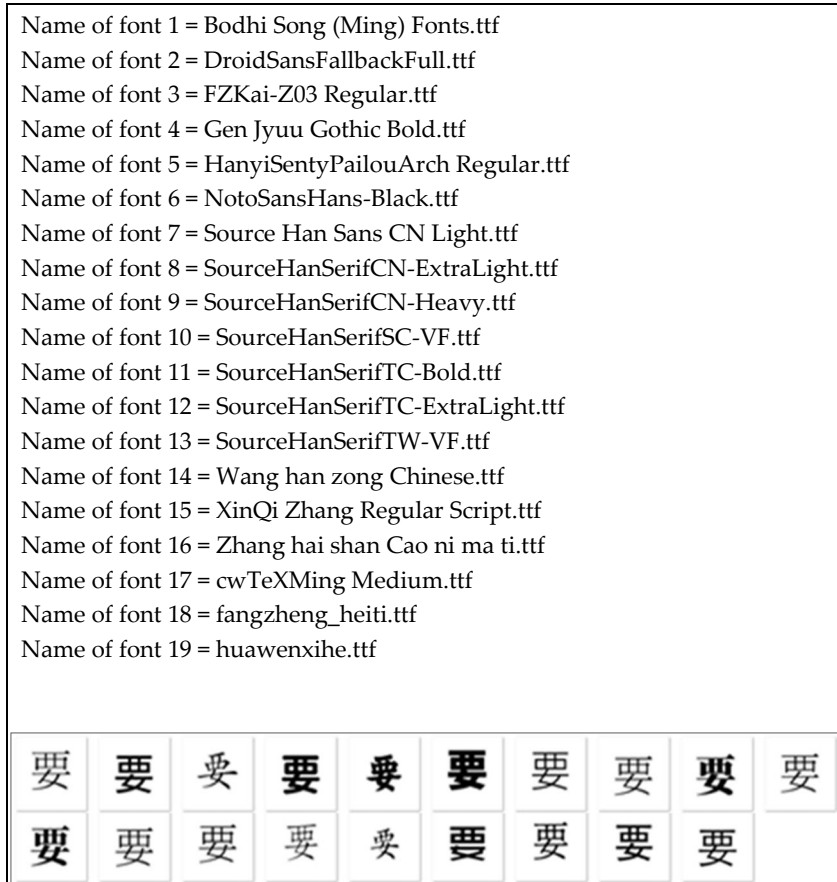

Name of font 1 = Bodhi Song (Ming) Fonts.ttf
Name of font 2 = DroidSansFallbackFull.ttf
Name of font 3 = FZKai-Z03 Regular.ttf
Name of font 4 = Gen Jyuu Gothic Bold.ttf
Name of font 5 = HanyiSentyPailouArch Regular.ttf
Name of font 6 = NotoSansHans-Black.ttf
Name of font 7 = Source Han Sans CN Light.ttf
Name of font 8 = SourceHanSerifCN-ExtraLight.ttf
Name of font 9 = SourceHanSerifCN-Heavy.ttf
Name of font 10 = SourceHanSerifSC-VF.ttf
Name of font 11 = SourceHanSerifTC-Bold.ttf
Name of font 12 = SourceHanSerifTC-ExtraLight.ttf
Name of font 13 = SourceHanSerifTW-VF.ttf
Name of font 14 = Wang han zong Chinese.ttf
Name of font 15 = XinQi Zhang Regular Script.ttf
Name of font 16 = Zhang hai shan Cao ni ma ti.ttf
Name of font 17 = cwTeXMing Medium.ttf
Name of font 18 = fangzheng_heiti.ttf
Name of font 19 = huawenxihe.ttf

**Figure 12.** Example characters in 19 different styles.

　　　Figure 13 shows examples of styled input images for random characters with one to nine components.

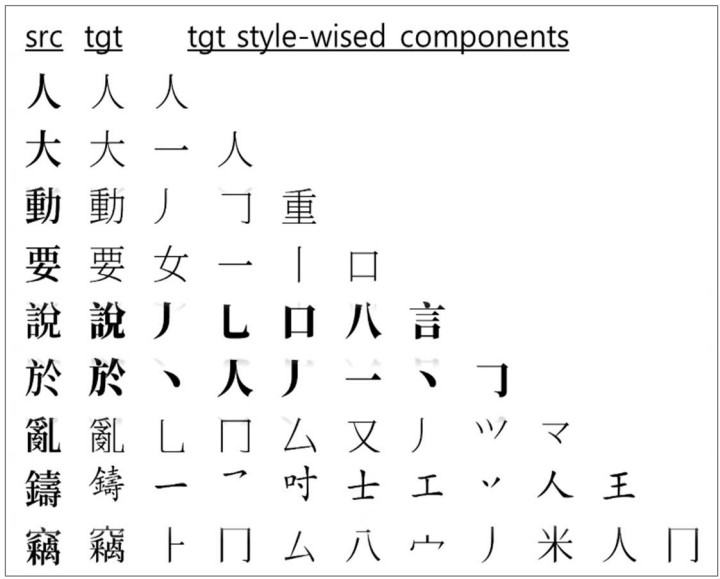

**Figure 13.** Input dataset sample by number of 1~9 components.

　　　Figure 14 shows the input representing a styled component for each font style for a random character.

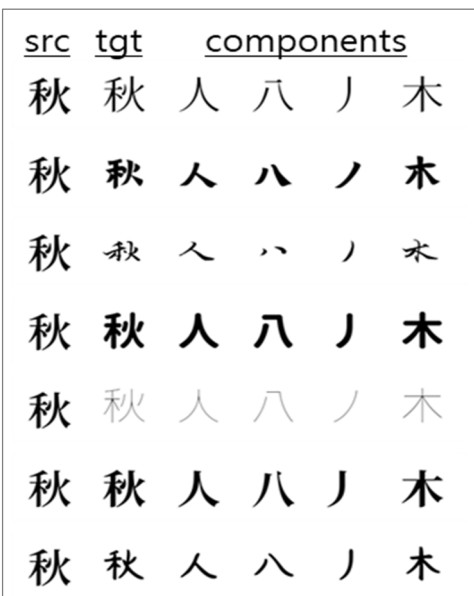

**Figure 14.** Examples of a character input dataset by font styles.

Figure 15 shows the overall data flow of the CCFont model. Figure 15a shows the data flow input to the CRC module. It demonstrates each Character-to-Radical-to-Components flow by decomposing and adding style information as an input style. This module generates content (Xc) and style (Xs), which are fed into encoders (Ec, Es) as inputs (Xc, Xs), converted to Zc and Zs, and then concatenated (Figure 15b). A new character set was generated through the decoder.

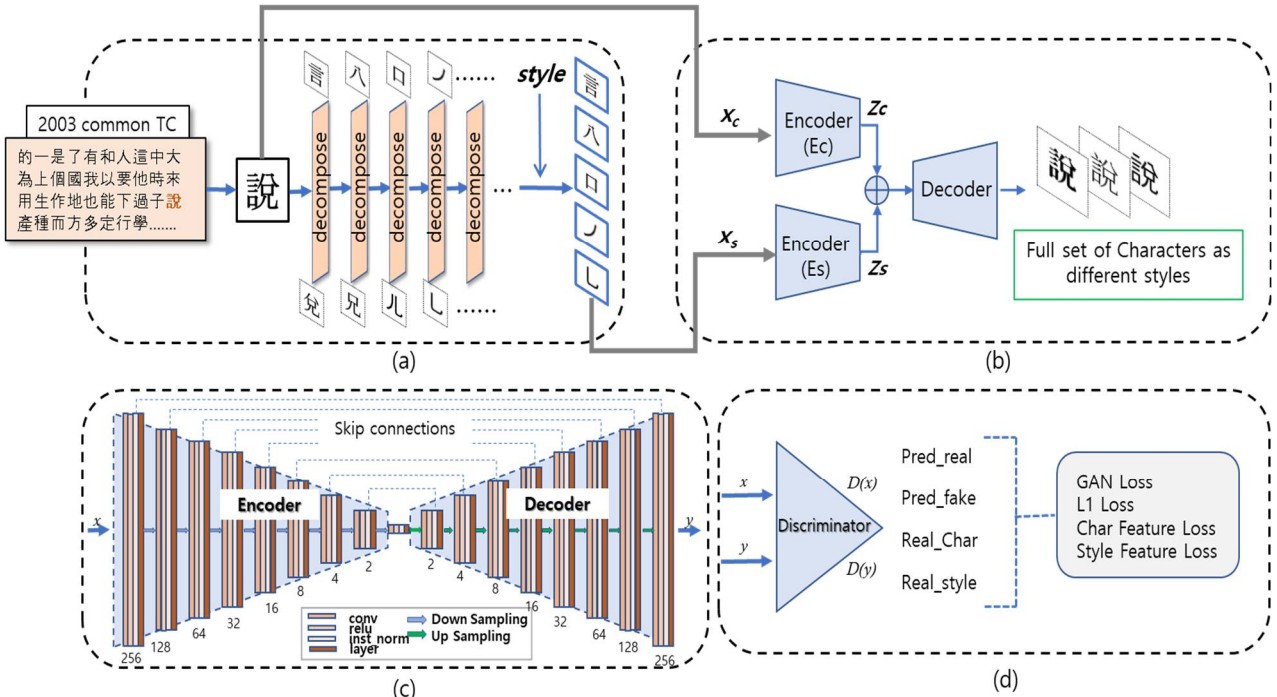

**Figure 15.** CCFont model data flow diagram. (**a**) CRC Module; (**b**) CCFont Generating Novel Characters Module; (**c**) CCFont Training Module (Generator) and (**d**) CCFont Training Module (Discriminator).

### 6.2. CCFont Model Training

The data generated by the CRC module were fed into the CCFont model training module and trained (Figure 15c,d). It took NVIDIA GTX-2080Ti GPU 12 GB-Memory size

23 h to train 306,794,188 parameters (10 epochs, 160,341 iterations for each epoch). Compared to a standard deep learning model trained on a high-performance GPU, the training time for this model was only 23 h, indicating that it is concise and resource-saving.

*6.3. Results of Font Generation (Generates Chinese and Multi-Lingual)*

The result of generating a new style of Chinese characters using the training module with the CCFont model is shown in Figure 16. The model was trained only on 2003 TC characters and successfully generated TC characters as well as SC characters that were seen for the first time. This result demonstrates that the model creates characters regardless of the component content by combining them. Through this, we verified that it is possible to create multiple languages using the TC character model, CCFont.

**Figure 16.** Sample output images of the TC characters generated.

Figure 16 shows the results of generating 100 TC Chinese characters and 10 font styles for the first time using the CCFont model, where new characters were created by accurately reflecting the target character's content and style.

Figure 17 shows the results of generating 300 SC Chinese characters and ten font styles. Despite being the first SC character, it has successfully generated new characters and demonstrated that it can be useful for TC-to-SC conversions.

Figure 18 compares the generated results of TC and SC characters for the same characters and font styles.

Figure 19a shows the result of generating 10 styles for the first 512 Korean characters seen for the first time, and Figure 19b shows the result of generating two styles for the first 13 Thai characters seen for the first time (zero-shot multi-lingual generation).

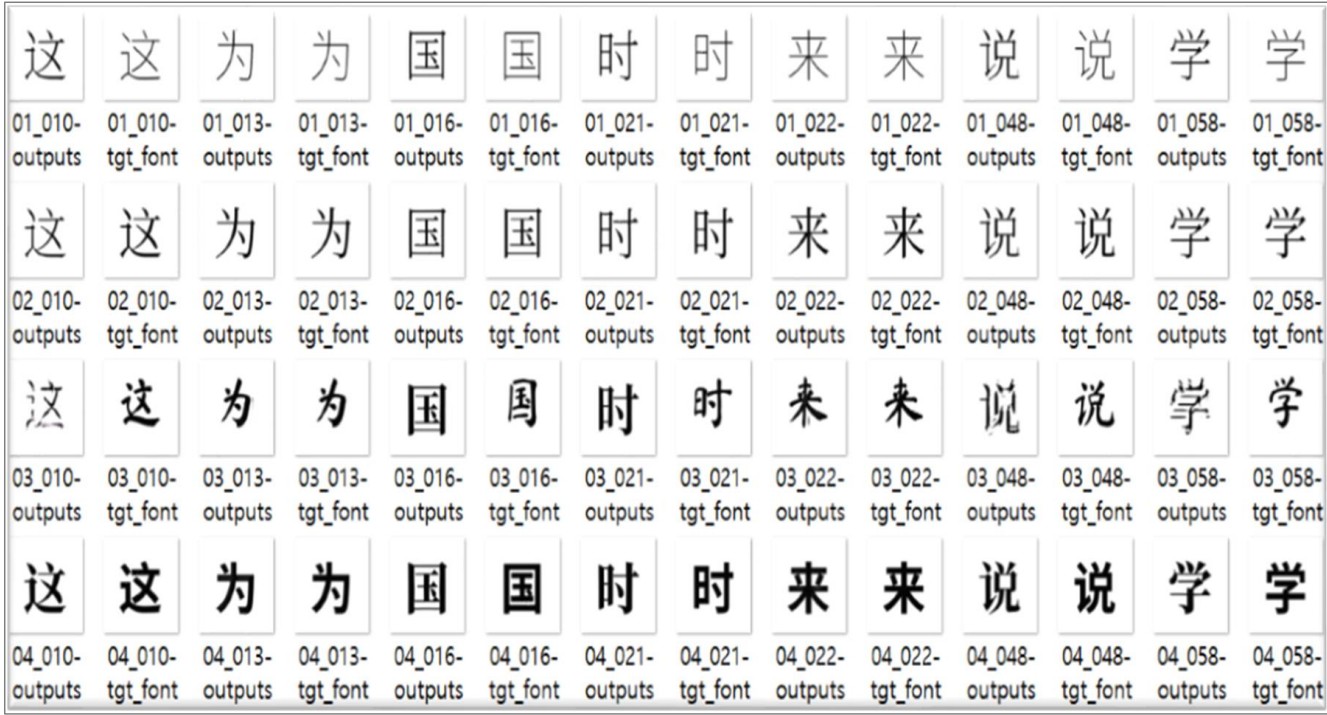

**Figure 17.** Sample output images of SC characters (left: output, right: target).

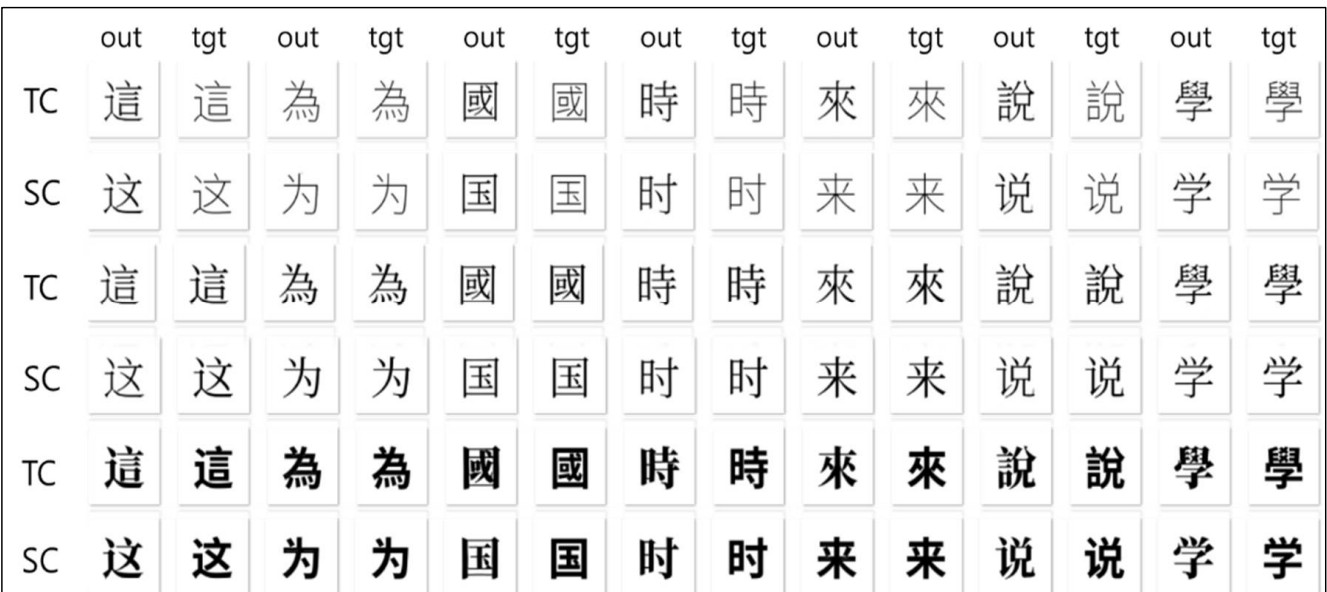

**Figure 18.** Generated output of TC and SC comparison by the CCFont model.

Hangul has 14 consonants and 10 vowels, and the sum of the initial (19 consonants) + middle (21 vowels) + final (28 consonants) forms 11,172 characters (19 × 21 × 28) [20]. Thai has 44 consonants, seven upper vowels, and nine highest, and four lower vowels (including cases without each) composed of initial (consonant) + vowel + final (consonant), making a total of 11,088 characters (44 × 7 × 9 × 4) (https://en.wikipedia.org/wiki/Thai_script, (accessed on 1 July 2022)).

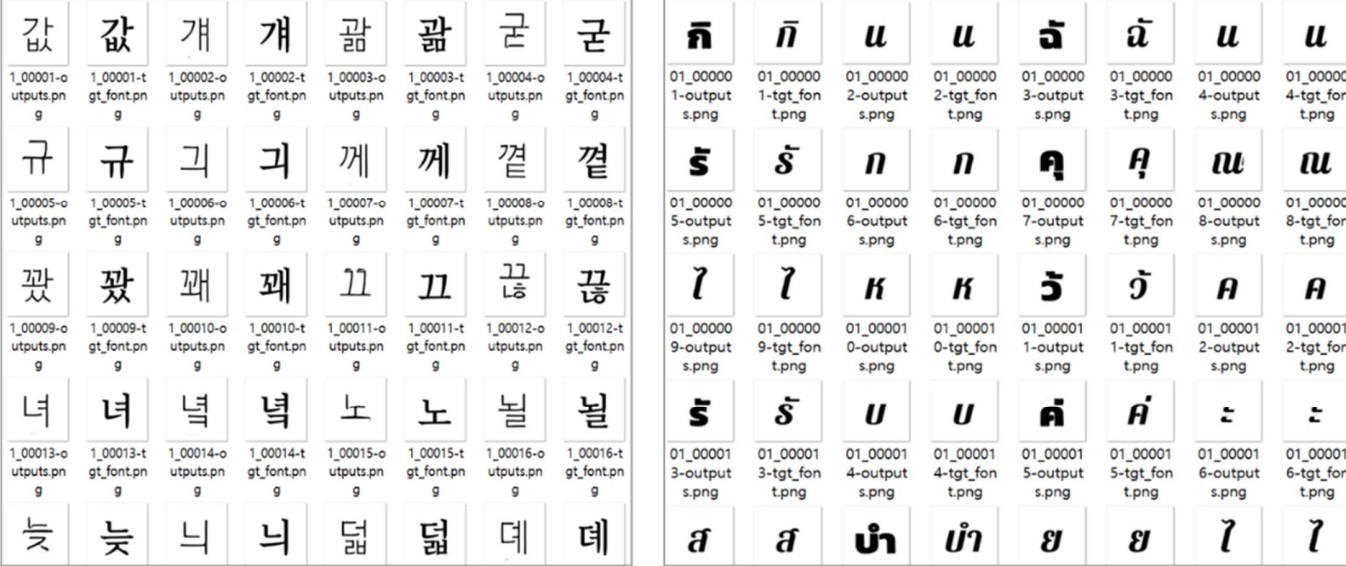

**Figure 19.** Sample images of Hangul and Thai characters generated by the CCFont Model.

In both Hangul and Thai, each character can be decomposed into components, resulting in the generation of each character by inputting the Chinese character component into a trained model. The content of the generated character is regenerated well, but style conversion is not performed because the fonts used for each language are different. The model trained only Chinese characters; however, if any language, including Korean and Thai, can be decomposed into its component parts, then the CCFont model can work on it.

## 7. Evaluation

### 7.1. Qualitative Evaluation

We selected the zi2zi and MX-Font models for comparison to verify the performance of CCFont. Each model has a different method of generating characters, but we ran the related models using the same character and font styles. Figure 20 depicts sample images for comparing the generation results of each model. Zi2zi and CCFont produced high-quality visual results, whereas MX-Font lacked style recognition. The reason MX-Font did not produce good results is because it requires more font styles to train.

| | | | | | | | | | |
|---|---|---|---|---|---|---|---|---|---|
| GT content | 的 | 一 | 下 | 共 | 看 | 各 | 新 | 前 | 中 | 明 |
| GT style | 的 | 一 | 下 | 共 | 看 | 各 | 新 | 前 | 中 | 明 |
| zi2zi | 的 | 一 | 下 | 共 | 看 | 各 | 新 | 前 | 中 | 明 |
| MX-Font | 的 | 二 | 下 | 共 | 看 | 各 | 新 | 前 | 中 | 明 |
| CCFont | 的 | 一 | 下 | 共 | 看 | 各 | 新 | 前 | 中 | 明 |

**Figure 20.** Example outputs by the proposed CCFont and the baselines.

The CCFont also had a failure case, which is considered to be a font-style problem. For example, as shown in Figure 21, failure cases appeared consistently only in font styles 3 and 5.

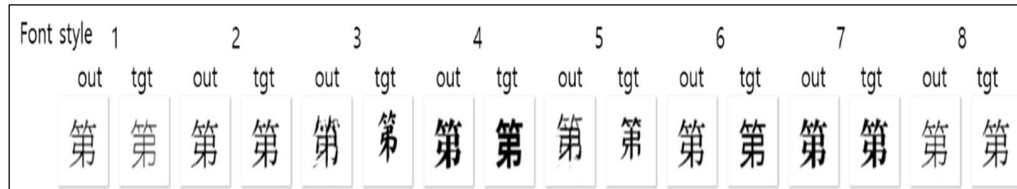

**Figure 21.** Failure examples (font styles 3 and 5).

Another reason arises from the component separation of images in Python programs. In the case of the character 向, the basic components show difficulties in regenerating the character because the Python program creates an image that is significantly different from the original character. In the character '向', the original image has (ノ) on top, but the Python separation program displays {'character': '向', 'compositions': ['ノ', '冂', '口']}. Because the separated image of 'ノ' is quite different from the shape of the original component, it causes difficulties in the recombination of character, as shown in Figure 22. The same phenomenon occurs in characters whose basic components provided by Python are different from the actual images, and failure cases occur depending on the font style compatibility. This can be inferred from the fact that the Thai vowel shape (กำ <=> ก, ◌ำ) is generated without failure. In addition, there is a problem with compatibility owing to the font style of different characters for each country.

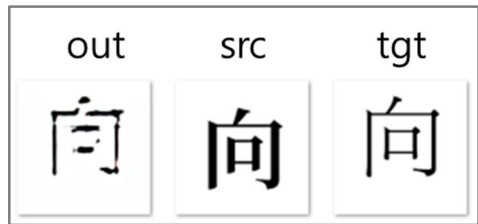

**Figure 22.** Failure examples of a character. Out, src, and tgt represent output, source, and target images, respectively.

### 7.2. Quantitative Evaluation

For quantitative comparison, we chose L1 and L2, Structural Similarity Index (SSIM) as our pixel-by-pixel difference metrics and FID (Frechet Inception Distance) scores [30] for distribution difference metrics. The results are presented in Table 2. Unseen TC 300 Chinese characters and fonts were used in each model to generate these results, and the values were calculated by comparing them with the ground truth. The zi2zi model requires fine-tuning, whereas the MX-Font and CCFont models do not. We also evaluated the CCFont model for both TC and SC characters.

**Table 2.** Quantitative comparative evaluation of the proposed method and the baselines.

| INDEX | L1-Loss ↓ | L2-Loss ↓ | SSIM ↑ | FID ↓ | Ref |
|:-----:|:---------:|:---------:|:------:|:-----:|:---:|
| **zi2zi** | 0.5104 | 0.4224 | **1.1619** | 104.8 | w/ finetuning |
| **MX-Font** | 0.5388 | 0.5360 | 0.3276 | 132.5 | w/o finetuning |
| **CCFont** | **0.2728** | **0.2689** | 0.8808 | **29.2** | w/o finetuning |

As shown in Table 2, the L1/L2 loss of the CCFont was the smallest. In contrast, the structural similarity of the zi2zi model was the highest, mainly because it requires extra fine-tuning, which is computationally expensive and time-consuming. The CCFont is slightly lower than the zi2zi model, and it shows that it is still a very high-quality result compared to the MX-Font. CCFont showed the best FID score, indicating that the lower the score, the closer it was to the original. Figure 23 shows a bar graph of the values in Table 2.

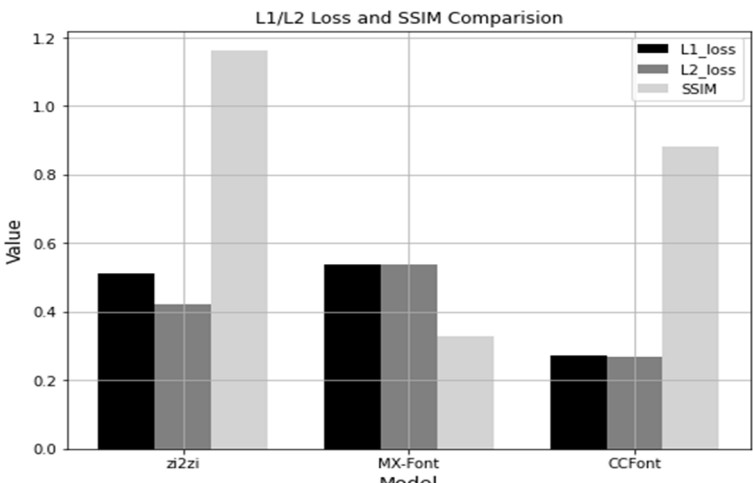

**Figure 23.** L1/L2 loss and SSIM comparison by models.

As shown in Figure 23, the L1/L2 loss value of CCFont was the best, and there was a significant difference between the models in terms of the degree of similarity. The result of zi2zi was the best, and the result of CCFont can be considered very good, as no additional training for unseen fonts was performed. MX-Font, which requires a large number of font styles, had a very low SSIM score because it has fewer font styles.

Figure 24 compares the FID scores for each model and shows that CCFont had the lowest scores, closest to the original image.

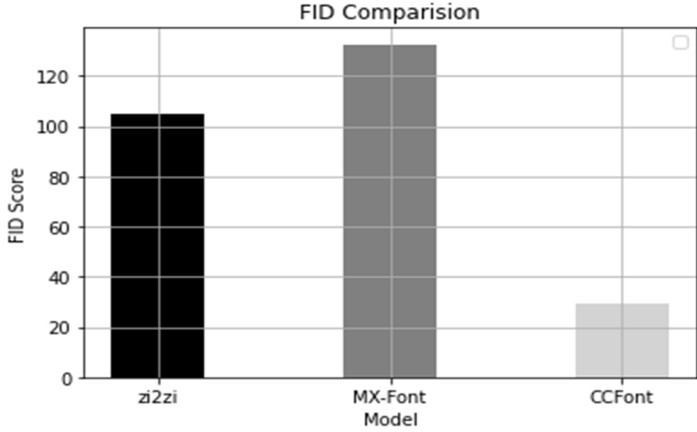

**Figure 24.** FID comparison of the three models.

## 8. Conclusions

In this study, we proposed a CCFont model that can convert high-quality Chinese character styles using Chinese character components and can generate Hangul and Thai with zero-shot without any additional training. CCFont is an automatic Chinese character generation model with a simple structure that conserves time and resources and produces high-quality results. To generate new Chinese fonts, it separates up to 17 Chinese characters and converts most Chinese characters into styles in a short amount of time. It is possible to generate not only TC Chinese characters but also SC Chinese, Korean, and Thai characters, and it can be applied to all fonts that can be separated into unattended components, including Japanese.

We also demonstrated that converting style images using component images can reduce failure cases, produce high-quality images, and save time and resources because of the concise structure of the model. In addition, we compared our results with those of relevant models to demonstrate better performance. In SSIM, the CCFont model was

slightly inferior to the finetuned zi2zi model but superior to the MX-Font model, and the FID score was the best of the three models.

In contrast to structural issues, some of the failure cases can be attributed to the incompatibility of different fonts between countries and the method of separating components. This aspect needs to be developed and improved in the near future.

**Author Contributions:** Conceptualization, Software, Writing, J.P.; methodology, A.U.H.; supervision, J.C. All authors have read and agreed to the published version of the manuscript.

**Funding:** This work was supported by Institute of Information & communications Technology Planning & Evaluation (IITP) grant funded by the Korea government (MSIT) (No. 2016-0-00166).

**Data Availability Statement:** Some or all data, models, or code that support the findings of this study are available from the corresponding author upon reasonable request.

**Conflicts of Interest:** The authors declare no conflict of interest.

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
