# Peer review of "CCFont: Component-Based Chinese Font Generation Model Using Generative Adversarial Networks (GANs)"

_applsci, doi:10.3390/app12168005_

Round 1

Reviewer 1 Report

The article presents a component-based Chinese font generation with generative adversarial networks. However, to improve the quality of the manuscript, the following issues should be addressed:

1-    The abstract should reflect the novelty of your work more clearly. i.e. How your proposed technique is different from similar reported works.

2-    To write a detailed introduction, the authors should include at least 5-6 more references.

3-    On what basis do you claim that the CCFont model you used is more efficient than other models used for similar purposes.

4-    Some Figures are blurry and not clear. i.e. Figure 18. If the Figures are taken from some reported work please cite them with permission.

5-    Table1 is empty. All the Figures and Tables should be center aligned.

6-    The Conclusions section should be crisp and to the point.

7-    The English writing of the manuscript can further be improved by a thorough review from an English expert.

Author Response

The article presents a component-based Chinese font generation with generative adversarial networks. However, to improve the quality of the manuscript, the following issues should be addressed:

1. The abstract should reflect the novelty of your work more clearly. i.e. How your proposed technique is different from similar reported works

=> We have rewritten the abstract to address the concerns of the reviewer.

2. To write a detailed introduction, the authors should include at least 5-6 more references.

=> We have modified the introduction and have added 12 more references in our updated manuscript.

3. On what basis do you claim that the CCFont model you used is more efficient than other models used for similar purposes.

=> Based on the following points we claim that the proposed model is efficient. We have added the following advantages at the end of introduction section as well. (line 92~104)

“The CCFont model proposed in this study has the following advantages and contributions:

  • Decomposing Chinese characters into components decreases the rendering difficulties significantly.
  • Component image conversion, which is much simpler than character shapes, significantly reduces the number of failure cases and enables high-quality character generation.
  • Up to 17 components could be generated, making it possible to generate almost all Chinese characters (99.99% in use).
  • Development of TC and SC Chinese character automatic generation / conversion model (CCFont) using Chinese character components
  • We experimentally demonstrated zero-shot font generation and generated Korean (Hangul) and Thai fonts using the CCFont model.”

4. Some Figures are blurry and not clear. i.e. Figure 18. If the Figures are taken from some reported work please cite them with permission.

=> We have added Figure 21 and changed the Figure 18 to Figure 22. All figures are created by our works except Figure 3 ~ 5 cited from Wikipedia as recommended by the reviewer.

5. Table1 is empty. All the Figures and Tables should be center aligned.

=> We apologize for such mistakes in our previous draft. We have updated our manuscript by carefully avoiding such mistakes. We have corrected and changed the tables and figures.

6. The Conclusions section should be crisp and to the point.

=> Following the comment, we have revised the Conclusion section.

Reviewer 2 Report

The authors propose a CCFont model (Components-based Chinese Font Generation Model) that automatically generates Chinese characters in various styles using a GAN architecture. The topic is novel but the paper must be improved extensively. My major concerns are as follows:

1) The motivation of the paper must be clarified in the paper. Moreover, the unique innovations and contributions of this paper should be further emphasized and clarified.

2) Authors must include another section and describe available handwritten databases in different languages such as IAM, RAMIS, SHIBR, Parzival, Washington, Saint Gall, Germana, Esposalles, and Rodrigo. Moreover, the limitations of these datasets must be discussed and explained clearly in the paper. 

3) Related work section must be definitely improved. It is poor. For instance, authors may consider including the following research papers:

CArDIS: A Swedish Historical Handwritten Character and Word Dataset

EMNIST: Extending MNIST to handwritten letters

Recognition of handwritten Chinese characters based on concept learning 

A new Arabic handwritten character recognition deep learning system (AHCR-DLS)

Recognition of Handwritten Chinese Characters Based on Concept Learning

TH-GAN: Generative Adversarial Network Based Transfer Learning for Historical Chinese Character Recognition

digitnet: a deep handwritten digit detection and recognition method using a new historical handwritten digit dataset

Improving GAN-Based Calligraphy Character Generation using Graph Matching

4) Fig. 10 must be further explained and discussed in the paper.

5) There is no information in Table 1. The authors must correct it.

6) Authors must definitely increase the quality of the figures in the paper. For instance, Figure 18 is very poor. Moreover, the tables and figures in the experimental section must be further discussed in the paper.

7) The authors must check the English in the paper, carefully.

Author Response

The authors propose a CCFont model (Components-based Chinese Font Generation Model) that automatically generates Chinese characters in various styles using a GAN architecture. The topic is novel but the paper must be improved extensively. My major concerns are as follows:

1. The motivation of the paper must be clarified in the paper. Moreover, the unique innovations and contributions of this paper should be further emphasized and clarified.

=> Following this comment, we have revised the abstract, introduction and have added advantages, contributions as in the paper (line 92~104)

2. Authors must include another section and describe available handwritten databases in different languages such as IAM, RAMIS, SHIBR, Parzival, Washington, Saint Gall, Germana, Esposalles, and Rodrigo. Moreover, the limitations of these datasets must be discussed and explained clearly in the paper. 

=> In this paper our main focus is on generating Chinese printed font characters therefore we have not added references related to handwritten databases.

3. Related work section must be definitely improved. It is poor. For instance, authors may consider including the following research papers:

a. CArDIS: A Swedish Historical Handwritten Character and Word Dataset

b. EMNIST: Extending MNIST to handwritten letters

c. Recognition of handwritten Chinese characters based on concept learning 

d. A new Arabic handwritten character recognition deep learning system (AHCR-DLS)

e. Recognition of Handwritten Chinese Characters Based on Concept Learning

f. TH-GAN: Generative Adversarial Network Based Transfer Learning for Historical Chinese Character Recognition

g. digitnet: a deep handwritten digit detection and recognition method using a new historical handwritten digit dataset

h. Improving GAN-Based Calligraphy Character Generation using Graph Matching

=> We have revised related work section and have added 12 more references to address the comment.

4. 10 must be further explained and discussed in the paper.

=> We have addressed this comment. The figures are renumbered as shown below.

5. There is no information in Table 1. The authors must correct it.

=> We apologize for such mistakes in our previous draft. We have updated our manuscript by carefully avoiding such mistakes. We have corrected and changed the tables and figures.

6. There is no information in Table 1. The authors must correct it.

=> We apologize for such mistakes in our previous draft. We have updated our manuscript by carefully avoiding such mistakes. We have corrected and changed the tables and figures.

7. The authors must check the English in the paper, carefully.

=> We have addressed this comment. We have used an English proofreading service to revise our final version of this paper.

Reviewer 3 Report

I have divided my comments into two parts.  The first part has comments about the technical content and what should be improved.  The second part is a line-by-line listing of grammatical or other issues I found that should be fixed.

Part 1:

You say many times why creating fonts is difficult but you never explain why this is difficult.  This is addressed later in the paper but should be at least partially described in the abstract and/or introduction.  You also state that the model can be used for other languages even though trained only on Chinese characters, and you demonstrate this at the end of the paper but it would be helpful to include some statement that you used your Chinese-trained model to do so, otherwise the reader must take your word on this until you prove it later.

I had an issue with understanding why this is a problem.  Since the fonts are already available, for instance in Python, why do you need to have a neural network learn how to generate a Chinese character?  This should be explained early in the paper. 

In the introduction, you reference 5-8 about the limitations but it would be best to give a partial explanation in the introduction. 

In section 2.1 you elude to DM-Font's limited performance but don't say what it was.  In the next paragraph, you start talking about your approach, but this section is supposed to be on related work, not yours.  Reference Zero-Shot (line 122) and Few-Shot (line 126).

Section 3:  you have 3.1 struck out, remove it.

On lines 252-253 you draw the conclusion that the model is applicable to other languages but you don't explain why this is so, do the other languages use the same strokes?  Beneath figure 9 you have in red (Figure 9 should be Table), make the change and remove this.

Section 4.1:  either explain cGAN or reference is.  The second paragraph ends with "Use the merged vector" but you don't say what the vectors are (presumably you are talking about Zc and Zs, which you introduce a couple of paragraphs later).

Section 4.2:  it would be helpful if you gave more of an intuitive explanation for these loss functions to go along with the formal descriptions.

Section 5.1:  Figure 13 should precede the first paragraph on page 12 since it was referenced earlier.  Figure 14c is hard to read.  You might think of breaking figure 14 into four separate figures so each part can be increased in size.

Section 5.2:  how many epochs did it take to do the training?  Someone might want to know this.

Table 5.1 is missing all of its data!

Part 2: (comments on various sentences that you should try to improve)

line 18:  "...is improved by converting using components" (drop "converting" I think)
line 25-26:  "This is a saving and efficient model."  (saving?, later you say time saving so maybe use "time saving and efficient" here)
line 28:  sentence ends with two periods
line 51-52:  "constituting" would probably be better stated as "consisting of"
line 55:  "with a high-quality of new style" is poorly stated
line 65:  "Deep Learning Learning" --> "Deep Learning"
line 71:  this paragraph ends with "to be.", apparently something you planned to fill in later?
lines 76-81:  each of these is stated in different verb tenses: "Development", "Provides", "All related"
line 93:  "an interesting task to challenge" is poorly stated
line 95:  "some characters has" --> "some characters have"
line 96/99:  you use "GAN" in line 96 but don't spell it out until line 99
lines 104-109:  this is a nightmare of a sentence and needs to be rewritten
line 112:  "Style information is the most important...along with content maintenance" --> then style information is not the most important if there is another, say "Style information and content maintenance are the most important..."
line 118:  you have "suggest" struck out, remove it in your final version
line 148:  "MX-Font, developed for multilingual by ..." something missing here, maybe "developed for multiple languages"
line 162:  Use "from" instead of "in" for the 1716 dictionary
lines 163-164:  this is all poorly worded
line 175:  stray ")" at the end of the second sentence (and no period)
line 180-181:  "components (radical/components)" -- having components twice is confusing, "the basic stroke is 6 types" should read "there are 6 basic types of strokes", and what is the difference between these 6 basic strokes and 41 total strokes?
line 190:  last sentence needs rewriting
line 193:  "is the left part is the radical" needs rewriting, maybe "in the left part"?
lines 219-220:  this sentence needs to be rewritten, the word "are" seems out of place
lines 234-235:  change "meaning" to "for"
line 239:  First sentence is not a sentence (no verb)
lines 242-243:  "the number of characters" is repeated
line 248:  "was used to learn" --> "were used for training", "up to 17 components were operated" --> "operated upon"? and missing a period
line 271-272:  "method for generating...is to accurately" -- something wrong here, maybe you should say that it is the goal of the method to accurately extract...
line 276:  ends with a stray "\"
line 285:  "Each encoder... 8 layers each", you can drop the second each, and add "and" before "except", change "convolution" to "convolutional" or "a convolution network"
line 287:  "stride=2," should be a period here
line 307:  "Each loss function" --> "The loss functions... are as follows."
line 336:  sentence ends with comma instead of period
lines 337-338:  very poorly written sentence
line 346:  "Python's"
line 437:  "the original image is (')" --> change "is" to "has"
line 449:  missing something, maybe say "A quantitative comparison is shown in table 1, displaying all calculations..."
line 451:  remove < > from around the Table 1 reference

Author Response

I have divided my comments into two parts.  The first part has comments about the technical content and what should be improved.  The second part is a line-by-line listing of grammatical or other issues I found that should be fixed.

Part 1:

 1. You say many times why creating fonts is difficult, but you never explain why this is difficult.  This is addressed later in the paper but should be at least partially described in the abstract and/or introduction. 

=> We have rewritten the abstract and introduction by adding why manual font design is difficult.

2. You also state that the model can be used for other languages even though trained only on Chinese characters, and you demonstrate this at the end of the paper but it would be helpful to include some statement that you used your Chinese-trained model to do so, otherwise the reader must take your word on this until you prove it later.

=> We have added the following lines in the introduction to address this concern.

Experiments demonstrate zero-shot font generation and generated Korean (Hangul) and Thai fonts using the CCFont model (line 103).

3. I had an issue with understanding why this is a problem.  Since the fonts are already available, for instance in Python, why do you need to have a neural network learn how to generate a Chinese character?  This should be explained early in the paper. 

=> Following explanation is added in the introduction to address this confusion: (line 36-43)

"When a font designer creates a new set of English fonts, they only need to design 52 uppercase and lowercase Roman characters. However, it is necessary to design 11,172 characters for Hangul and 70,000 to 100,000 characters for Chinese, which is labor-intensive and practically impossible. These labor-intensive tasks take 700 days for Hangul and more than 10 years for Chinese, and it is almost impossible to design them in the same style even if a designer makes one character every 30 minutes and works 8 hours a day [1].”

4. In the introduction, you reference 5-8 about the limitations but it would be best to give a partial explanation in the introduction. 

=> We have added Figure 1 and explained the difficulties of manual font design in our updated introduction.

5. In section 2.1 you elude to DM-Font's limited performance but don't say what it was.  In the next paragraph, you start talking about your approach, but this section is supposed to be on related work, not yours.  Reference Zero-Shot (line 122) and Few-Shot (line 126).

=> We have rewritten the related works section in order to make sure about the sequence of the related works. We have further added two subsections to clearly discuss the related literature.

6. Section 3:  you have 3.1 struck out, remove it.

=> We have revised such alignment issues in our updated manuscript.

7. On lines 252-253 you draw the conclusion that the model is applicable to other languages but you don't explain why this is so, do the other languages use the same strokes?  Beneath figure 9 you have in red (Figure 9 should be Table), make the change and remove this.

=> In section 4, we demonstrate how our proposed model can be adopted for other component-based languages. Mainly we have explained this from Line 290–301. We have fixed the Figure 9 problem as mentioned in the comment but because of the reordering of the figures and tables, The original Figure 9 became Figure 8 in the revised version.

8. Section 4.1:  either explain cGAN or reference is.  The second paragraph ends with "Use the merged vector" but you don't say what the vectors are (presumably you are talking about Zc and Zs, which you introduce a couple of paragraphs later).

=> We have added the following explanations

“The resulting latent vectors from the 2 encoders, Zc and Zs, are merged” (line 322)

9. Section 4.2:  it would be helpful if you gave more of an intuitive explanation for these loss functions to go along with the formal descriptions.

=> We have completely revised our Section of Loss functions the new Section number is 5.2, and all the losses are clearly explained with formulations.

10. Section 5.1:  Figure 13 should precede the first paragraph on page 12 since it was referenced earlier.  Figure 14c is hard to read.  You might think of breaking figure 14 into four separate figures so each part can be increased in size.

=> We have rearranged and renumbered all the Figures and Tables of our revised manuscript.

11. Section 5.2:  how many epochs did it take to do the training?  Someone might want to know this.

=> We have added this explanation in the manuscript: 10 epochs, 160,341 iterations for each epoch. (line 421)

12. Table 5.1 is missing all of its data!

=> We apologize for such mistakes in our previous version. We have carefully fixed all such mistakes in the revised manuscript.

Part 2: (comments on various sentences that you should try to improve)

line 18:  "...is improved by converting using components" (drop "converting" I think)
line 25-26:  "This is a saving and efficient model."  (saving?, later you say time saving so maybe use "time saving and efficient" here)
line 28:  sentence ends with two periods
line 51-52:  "constituting" would probably be better stated as "consisting of"
line 55:  "with a high-quality of new style" is poorly stated
line 65:  "Deep Learning Learning" --> "Deep Learning"
line 71:  this paragraph ends with "to be.", apparently something you planned to fill in later?
lines 76-81:  each of these is stated in different verb tenses: "Development", "Provides", "All related"
line 93:  "an interesting task to challenge" is poorly stated
line 95:  "some characters has" --> "some characters have"
line 96/99:  you use "GAN" in line 96 but don't spell it out until line 99
lines 104-109:  this is a nightmare of a sentence and needs to be rewritten
line 112:  "Style information is the most important...along with content maintenance" --> then style information is not the most important if there is another, say "Style information and content maintenance are the most important..."
line 118:  you have "suggest" struck out, remove it in your final version
line 148:  "MX-Font, developed for multilingual by ..." something missing here, maybe "developed for multiple languages"
line 162:  Use "from" instead of "in" for the 1716 dictionary
lines 163-164:  this is all poorly worded
line 175:  stray ")" at the end of the second sentence (and no period)
line 180-181:  "components (radical/components)" -- having components twice is confusing, "the basic stroke is 6 types" should read "there are 6 basic types of strokes", and what is the difference between these 6 basic strokes and 41 total strokes?
line 190:  last sentence needs rewriting
line 193:  "is the left part is the radical" needs rewriting, maybe "in the left part"?
lines 219-220:  this sentence needs to be rewritten, the word "are" seems out of place
lines 234-235:  change "meaning" to "for"
line 239:  First sentence is not a sentence (no verb)
lines 242-243:  "the number of characters" is repeated
line 248:  "was used to learn" --> "were used for training", "up to 17 components were operated" --> "operated upon"? and missing a period
line 271-272:  "method for generating...is to accurately" -- something wrong here, maybe you should say that it is the goal of the method to accurately extract...
line 276:  ends with a stray "\"
line 285:  "Each encoder... 8 layers each", you can drop the second each, and add "and" before "except", change "convolution" to "convolutional" or "a convolution network"
line 287:  "stride=2," should be a period here
line 307:  "Each loss function" --> "The loss functions... are as follows."
line 336:  sentence ends with comma instead of period
lines 337-338:  very poorly written sentence
line 346:  "Python's"
line 437:  "the original image is (')" --> change "is" to "has"
line 449:  missing something, maybe say "A quantitative comparison is shown in table 1, displaying all calculations..."
line 451:  remove < > from around the Table 1 reference

=> All incorrect sentences/parts have been corrected. More specifically, we have received the professional proof-reading service to fix the grammatical and other English related mistakes in the previous manuscript.

Round 2

Reviewer 1 Report

No more comments from the reviewer.

Author Response

We sincerely appreciate for the reviewer’s comments to improve the quality of our paper. 

And we received the professional English proofreading services on July 29.
Encloes is the proof of "The Editing Certificate." 

Reviewer 2 Report

The authors should properly address my previous comments in the paper to enhance the quality of the paper. In addition, the authors should address the following comments:

1) The authors use conditional GAN, which is necessary to explain with details in the paper in an additional section.

2) Figures 3 and 14 must be explained with more details. What are the src, tgt etc. mentioned in these figures?

3) To understand the performance of the proposed method, the authors must perform another GAN model and include the output results of the compared model in Figures 16, 17, and 18. The qualitative results should also be discussed in detail.

4) The text is under Table 2. So, it cannot be readable. The authors should correct this.

5) Caption of Figure 23 must be corrected.

6) The authors should remove the following references from the journal (which are decreasing the scientific sound):

32. https://pypi.org/project/cjkradlib/ 599 (this page doesn't work)

33. https://en.wikipedia.org/wiki/Thai_script

22. https://en.wikipedia.org/wiki/Chinese_characters 583

Author Response

The authors should properly address my previous comments in the paper to enhance the quality of the paper. In addition, the authors should address the following comments:

=> We sincerely appreciate for the reviewer’s comments to improve the quality of our paper. In this new revised version, many figures and tables have been corrected to clarify their meaning. We also did our best to carefully follow all comments as follows:

1) The authors use conditional GAN, which is necessary to explain with details in the paper in an additional section.

=> Conditional GAN (cGAN) has the advantage of controlling the output compared with the vanilla GAN and its been widely used since it was published in 2014. However, as cGAN became popular and widely used, cGAN replaced GAN and was generalized to various computer vision tasks. Recently, cGAN is used as a basic algorithm representing GAN without any special explanation. The figure shows the difference between GAN and cGAN. (The figure is included in the attached file.)

=> Following this comment, we have revised the related works section and have added few lines about the conditional GAN as recommended by the reviewer. Please refer to Section ‘2.1 Many-shot Font generation methods’ in the paper (line 119-122) 

lines (209-211) in the paper
The conditional GAN is the extension of vanilla GAN, where the image is generated with some condition c, i.e., c can be a class label or an image (our case). The condition c is added to both the generator and discriminator for the parameters intended to be controlled.

2) Figures 3 and 14 must be explained with more details. What are the src, tgt etc. mentioned in these figures?

=> We have explained the details of Figure 3 in lines (209-211) and (213-215).

lines (209-211) in the paper
Chinese characters can be divided into strokes and their components (radicals). A stroke is the smallest unit comprising a character. There are six types of basic strokes, as shown in Figure 3, and a total of 41 types of strokes.

lines (213-215) in the paper
The strokes form part of a character but have no meaning. The strokes are useful for writing and identifying characters, but they are too small to be used as elements to form an image of a character.

=> The terms of ‘src’ and ‘tgt’ first appear in Figure 9 in the paper. We have described these terms in line (266-269).

lines (266-269) in the paper
The sample input image to the CCFont model is shown in Figure 9, where src represents the source image, tgt represents the target image, and all the basic components of tgt image representing the target font style (tgt style-wised components) are demonstrated. 

3) To understand the performance of the proposed method, the authors must perform another GAN model and include the output results of the compared model in Figures 16, 17, and 18. The qualitative results should also be discussed in detail.

=> Figures 16, 17, and 18 are about demonstrating the outputs of only our proposed CCFont model. For a fair comparison of our proposed model against other GAN models of zi2zi and MX-Font, we have conducted experiments based on visual quality and quantitative metrics in Section 7. The results are demonstrated in Figures 20, 23, 24, and Table 2, respectively.

lines (456-463) in the paper
7. Evaluation
7.1 Qualitative evaluation
We selected the zi2zi and MX-Font models for comparison to verify the performance of CCFont. Each model has a different method of generating characters, but we ran the related models using the same character and font styles. Figure 20 depicts sample images for comparing the generation results of each model. Zi2zi and CCFont produced high-quality visual results, whereas MX-Font lacked style recognition. The reason MX-Font did not produce good results is because it requires more font styles to train.

lines (483-509) in the paper
7.2 Quantitative evaluation
For quantitative comparison, we choose L1 and L2, Structural Similarity Index (SSIM) as our pixel-by-pixel difference metrics, and FID (Frechet Inception Distance) scores [30] for distribution difference metrics. The results are presented in Table 2. Unseen TC 300 Chinese characters and fonts were used in each model to generate these results, and the values were calculated by comparing them with the ground truth. The zi2zi model requires fine-tuning, whereas the MX-Font and CCFont models do not. We also evaluated the CCFont model for both TC and SC characters.

As shown in Table 2, the L1/L2 loss of the CCFont was the smallest. In contrast, the structural similarity of the zi2zi model was the highest, mainly because it requires extra fine-tuning, which is computationally expensive and time-consuming. The CCFont is slightly lower than the zi2zi model, and it shows that it is still a very high-quality result compared to the MX-Font. CCFont showed the best FID score, indicating that the lower the score, the closer it was to the original. Figure 23 shows a bar graph of the values in Table 2.

As shown in Figure 23, the L1/L2 loss value of CCFont was the best, and there was a significant difference between the models in terms of the degree of similarity. The result of zi2zi was the best, and the result of CCFont can be considered very good, as no additional training for unseen fonts was performed. MX-Font, which requires a large number of font styles, has a very low SSIM score because it has fewer font styles.

Figure 24 compares the FID scores for each model and shows that CCFont has the lowest scores, closest to the original image.

4) The text is under Table 2. So, it cannot be readable. The authors should correct this.

=> We have revised and apologize for such mistakes. The modification is shown in the previous page (Please see answer of (3). And Table 2 is modified.)

5) Caption of Figure 23 must be corrected.

  • We have addressed this comment and changed the title of Figure 23 as “L1/L2 loss and SSIM comparison”. (Please see Figure 23 in the attached file or in the revised paper.)

6) The authors should remove the following references from the journal (which are decreasing the scientific sound):

  1. https://pypi.org/project/cjkradlib/ 599 (this page doesn't work)
  2. https://en.wikipedia.org/wiki/Thai_script
  3. https://en.wikipedia.org/wiki/Chinese_characters 583

=> We have addressed this comment and removed these from the reference section. Instead, we have added these as footnotes. (Please the modification in the attached file.)

7) We receive the professional English editing service. Enclosed is the editing certificate. (The certificate is enclosed in the reviewer2 response file.)
